# Sustainability in Intensive Aquaculture—Profitability of Common Carp (*Cyprinus carpio*) Production in Recirculating Aquaculture Systems Based on a Hungarian Case Study

**DOI:** 10.3390/ani15071055

**Published:** 2025-04-05

**Authors:** Laura Mihály-Karnai, Milán Fehér, Péter Bársony, István Szűcs, Tamás Mihály, Dániel Fróna, László Szőllősi

**Affiliations:** 1Institute of Economics, Faculty of Economics and Business, University of Debrecen, Böszörményi Str. 138, 4032 Debrecen, Hungary; karnai.laura@econ.unideb.hu (L.M.-K.); szucs.istvan@econ.unideb.hu (I.S.); szollosi.laszlo@econ.unideb.hu (L.S.); 2Institute of Animal Science, Biotechnology and Nature Conservation, Faculty of Agricultural and Food Sciences and Environmental Management, University of Debrecen, Böszörményi Str. 138, 4032 Debrecen, Hungary; feherm@agr.unideb.hu (M.F.); barsonp@agr.unideb.hu (P.B.); 3Institute of Marketing and Commerce, Faculty of Economics and Business, University of Debrecen, Böszörményi Str. 138, 4032 Debrecen, Hungary; mihaly.tamas@econ.unideb.hu

**Keywords:** aquaculture, sustainability, sustainable carp production, recirculating aquaculture systems, common carp, costs, profitability

## Abstract

This study examines the intensive aquaculture of common carp (*Cyprinus carpio*) in Hungary, focusing on the economic feasibility of recirculating aquaculture systems (RASs). Using a deterministic model, it analyzes production costs, profitability, and factors influencing economic efficiency, such as feed costs, electricity consumption, and market sale prices. The research breaks down the production process into three main phases—pre-rearing, post-rearing, and market-sized fish production—identifying high initial investments and operational costs as major challenges. However, it also highlights opportunities for optimizing feed management and energy efficiency to improve sustainability and profitability. The findings are crucial for integrating economic, environmental, and technological aspects to advance sustainable aquaculture practices.

## 1. Introduction

Aquaculture significantly contributes to global food production, supports the livelihoods of millions worldwide, and plays a crucial role in international trade while offering potential solutions to food security challenges [1,2]. However, its rapid expansion requires the careful consideration of sustainable development principles to reduce environmental impacts and ensure long-term viability [2]. Sustainable aquaculture integrates production efficiency with environmental responsibility, economic feasibility, and social considerations to mitigate long-term industry risks. This approach involves implementing practices that minimize ecological impact, such as reducing water pollution, conserving biodiversity, and optimizing resource utilization [1]. Aquaculture can have a smaller ecological footprint than terrestrial protein sources; however, unsustainable practices can lead to significant risks, including habitat destruction, water pollution, and biodiversity loss. To reduce these negative impacts, it is important to implement best management practices and follow programs that promote responsible production [3,4]. Sustainable aquaculture socially supports community development by creating jobs and enhancing food security [5]. So, aquaculture with the principles of sustainable development is essential for its long-term viability. Implementing sustainable aquaculture practices can contribute to global food security while mitigating negative environmental impacts.

### 1.1. State of Global Fisheries

The global consumption of aquatic food has significantly increased over the decades, with an annual growth rate that surpasses that of the global population. The primary factors driving the continued growth in per capita consumption include increased supply, the preservation and distribution of technological advancements, shifting consumer preferences, and rising incomes [6].

Freshwater fish lead aquaculture production at 83%, followed by diadromous fish at 79%. These high percentages demonstrate the successful implementation of farming techniques in these sectors. Mollusks and miscellaneous aquatic animals also show a strong aquaculture presence at 75% and 72%, respectively. Marine fish present a striking contrast, with capture fisheries accounting for 95% of production. This dominance reflects the traditional and continued importance of wild-catch methods in marine environments. Crustaceans show a more balanced distribution, with capture fisheries representing 32% of production (Figure 1) [6].

The total production volume, indicated by the line in Figure 1, reveals that marine fish achieve the highest output at approximately 70 million tonnes. Freshwater fish maintain substantial production at around 60 million tonnes, while crustaceans and mollusks show moderate volumes between 20 and 40 million tonnes. The miscellaneous aquatic animals’ category records the lowest volume at about 5 million tonnes [6].

The number of species harvested has varied significantly across different regions. Until the late 1970s, finfish comprised about 90% of aquatic animal production. However, by 2022, this figure had decreased to 75% due to the rise in aquaculture production, which has increased the share of mollusks and crustaceans. In 2022, marine fish accounted for 50% of the total aquatic animal production, while freshwater fish represented 44% of the total finfish and 33% of the overall aquatic animal production (Figure 1) [6].

The leading group of species produced in 2022 were carp, barbels, and other cyprinids (*Cyprinidae*), accounting for 18% of aquaculture production, followed by various freshwater species (11%) and clupeiforms (*Clupeiformes*), such as herring (*Clupea* spp.), sardines (*Sardinella* spp.), and anchovies (*Engraulis* spp.) (10%). At the species level, whiteleg shrimp (*Penaeus vannamei*) was the most produced species in 2022, at 6.8 million tonnes, closely followed by cupped oysters (*Crassostrea* spp.) at 6.2 million tonnes, grass carp (*Ctenopharyngodon idella*, also known as white amur) at 6.2 million tonnes, Nile tilapia (*Oreochromis niloticus*) at 5.3 million tonnes, silver carp (*Hypophthalmichthys molitrix*) at 5.1 million tonnes and Peruvian anchovy (*Engraulis ringens*) at 4.9 million tonnes. It is worth noting that, in 2022, aquaculture was the primary source of production for the five largest aquatic animal species and eight of the ten largest aquatic animal species (Figure 2) [6].

### 1.2. Advancements in Sustainable Practices

Aquaculture is a rapidly growing sector in food production, leading to increased intensity in the industry and a larger workforce. However, rapid industry expansion, when not properly regulated, has been associated with environmental challenges, such as water pollution and habitat degradation. Inappropriate aquaculture can harm genetic diversity, disrupt ecosystems, affect human health, deplete natural resources, and degrade water quality. To address these concerns, sustainable aquaculture practices are essential for minimizing the risks associated with this industry and ensuring food safety for human consumption [7]. The foundation of sustainable aquaculture is based on balancing environmental integrity, economic viability, and social equity [8]. According to Engle and van Senten [9], these principles ensure that aquaculture systems operate profitably while maintaining ecological balance and enhancing community livelihoods.

Innovations in technology have made sustainable aquaculture much more advanced. Recirculating aquaculture systems (RASs) are revolutionary because effective water recycling systems are used to reduce waste discharge and water demand [10]. Selective breeding programs have enhanced disease resistance and growth rates, reducing the reliance on chemical interventions [11]. The integration of seaweed farming into aquaculture has gained attention due to its environmental benefits. Seaweed serves as a biofilter by absorbing excess nutrients and helps reduce water pollution [12,13]. RASs, integrated multi-trophic aquaculture (IMTA), and biofloc technology (BFT) are mitigation techniques that use lower volumes of water, reduce waste, and enhance water quality [14,15]. Substituting feed elements like plant-based proteins and insect meal reduces reliance on wild fish sources, supporting ecological sustainability by minimizing nitrogen excretion into water systems while maintaining growth performance and animal health. This highlights the need for innovative feed methods to lessen dependency on forage fish and avoid ecological constraints [16,17,18].

The FAO’s *Blue Transformation* roadmap (2022–2030) presents a strategic vision for changing aquatic food systems to help achieve the sustainable development goals (SDGs) and address global issues like poverty, hunger, malnutrition, and environmental degradation. The goal of aquaculture’s sustainable growth is to increase production by 35% by 2030, with the help of cutting-edge technologies and fair resource distribution [19].

In Hungarian pond farms, common carp (*Cyprinus carpio*) production is currently typically carried out in a three-year cycle. This is the prevalent, classic production strategy, although there are exceptions and efforts to increase intensity [20,21]. The three-year-cycle production strategy takes an extremely long time. Recirculation systems [22], cage fish farming systems [23], and “pond-to-pond” systems may also be suitable for common carp farming. Shortening the production cycle has several advantages, including reduced production risks (fish diseases, over-wintering, fish kills, bird damage, and theft). It is therefore worth looking into the possibilities of shortening the growing season. Based on our current knowledge, several solutions are available in practice. Hatchery rearing is one of the most critical periods of production, with an average mortality rate of 40–60%. In unfavorable conditions, this can rise to 90–95%. This short but highly sensitive period of only 3–4 weeks often determines the yield for the whole year and has a significant impact on production in the following two years [24]. It is therefore worth considering intensive rearing under controlled conditions as an option.

Increasing consumer demand has stimulated the development of industrial fish farming, which involves fish production in relatively small areas, with a high degree of mechanization that maintains water quality and temperature in the optimum range, and at maximum yield. The most prominent forms of intensive fish farming include RASs [25], which operate under closed, precisely controlled conditions, allowing the production of large quantities of fish flesh of constant quality in a short time, independent of external climatic factors [26,27]. RASs have low water and land requirements but require significant initial investment, high operating costs, and a high level of expertise; however, they are capable of producing a wide range of fish species and provide a continuous supply of fish [28]. This allows for a safer and more efficient form of common carp production, with a shorter production cycle [29,30,31,32,33]. Figure 3 illustrates a typical RAS setup.

The objective of this study is to provide a comprehensive analysis of production parameters, cost and income relationships, and production efficiency in an intensive closed-system common carp farming experiment. Achievement of this objective is demonstrated by a deterministic economic model and comprehensive sensitivity analysis. The main research questions related to the objective are the following:(1)What are the technological characteristics of common carp production in RASs?(2)Which production parameters (net yield, call, specific feed consumption, etc.) characterize carp production in RASs?(3)What was the profitability of carp production in RASs in Hungary in the economic environment of 2024?(4)What are the factors that most influence profitability, and what are considered to be the critical points of the technology?(5)How do changes in key production parameters affect economic indicators?

## 2. Materials and Methods

### 2.1. Data Collection

The data for this study were obtained from an intensive system trial conducted in Hungary. A semi-industrial scale experiment on common carp farming, breeding, and rearing was carried out in a RAS, which also served as the source of data collection. The experiment took place at the Fish Biology Laboratory of the Faculty of Agriculture and Food Science and Environmental Management, University of Debrecen (Debrecen, Hungary), between 2020 and 2021. The fish stock used for the experiment was produced as part of an off-season induced common carp propagation program at the hatchery of Bocskai Halászati Kft. (Hajdúszoboszló, Hungary). Using a RAS, common carp farming was conducted in a controlled and optimized environment, where the fish were reared under artificial conditions over approximately 12 months.

The semi-industrial scale experiment was not repeated due to its long duration and high costs. Regarding data robustness, the experiment involved a large fish population compared to classical controlled studies designed to assess single-factor effects. This is reflected in the stocking density data across production phases. The collected dataset provided a substantial empirical basis for the econometric calculations conducted. Furthermore, unlike conventional short-term trials that typically last 42–56 days, this study covered a 12-month production cycle, offering a more comprehensive assessment of economic feasibility.

Data collection encompassed technological, operational, and financial data. Technological and operational data were collected from detailed plant records in different phases of the production process. These records included rearing conditions (such as water volume, water usage, water exchange or replacement, electricity usage, labor requirements, and water temperature) and production parameters (such as phase duration, stocking density, individual body weight, survival rate, mortality, grading, body weight at harvest, number of test catches, number of feeding days, and the type and amount of feed used). Based on the production data, indicators were defined for each production stage and aggregated for the entire production process:Fish loss (%) = 100 − (number of live fish harvested (pc)/number of fish released (pc)) × 100;Average body weight at harvest (kg/pc) = amount of live weight harvested (kg)/number of fish harvested (pc);Gross yield (kg) = amount of live weight harvested (kg) − amount of live weight released (kg);Specific gross yield of carp (kg/m^3^) = gross yield (kg)/production capacity (m^3^);Feed conversion ratio (FCR) (kg/kg) = F/(W_f_ − W_i_), where F is the feed consumed during the experiment (kg), W_f_ is the final body weight (kg), and W_i_ is the initial body weight (kg);Stocking density (fish/m^3^) = number of individuals stocked (fish)/production capacity (m^3^).

In this experiment, the production process was divided into three phases (Table 1), during which market-sized common carp were produced in approximately one year. Throughout all three phases, the experimental stock was reared in a closed RAS under controlled conditions, equipped with mechanical and aerated biofilters as well as UV lamps.

Phase 1 involved larval rearing, during which a 1 g larva developed from self-reared tender larvae over a 60-day pre-rearing period. Following the February breeding trial, 3100 larvae were stocked into a single 400 L plastic circular tank. The total capacity of the RAS for larval pre-rearing, based on the usable water volume, was 0.35 m^3^. During the first two weeks, the common carp larvae were fed live food, specifically leeches (*Artemia* spp.). After two days of pre-rearing, they were transitioned to commercially available dry feed (BioMar Inicio Plus G; 0.5; 0.8, BioMar Group, Aarhus, Denmark). Oxygen saturation was maintained above 80% using atmospheric aeration, while the temperature was stabilized at 22 ± 1.5 °C. The lighting cycle consisted of 12 h light and 12 h dark periods. The water temperature and dissolved oxygen levels were monitored daily with a HACH HQ30d portable meter (HACH Co., Loveland, CO, USA). NO_2_^−^, NO_3_^−^, and NH_4_^+^ concentrations were measured with a HACH Lange DR/3900 spectrophotometer (HACH Co., Loveland, CO, USA). With an average mortality rate of 60%, approximately 1800 fry per larval rearing unit had survived at the end of the 8.5-week pre-rearing period and were transferred to the next phase. The average individual weight gain during pre-rearing was 0.02 g/day, resulting in an expected 5.6 kg of pre-reared fry per production cycle.

Phase 2 began when pre-reared fry, weighing 1.05 g, were transferred and reared for 90 days, reaching an individual weight of 120 g. In this phase, which started in April, the fry were transferred from the pre-rearing phase to the post-rearing phase. Rearing was conducted in a separate recirculation system consisting of twelve 1 m^3^ plastic circular tanks with a total usable water volume of 12 m^3^ [35]. Fish weights were measured using a VWR SE 422 digital balance with 0.01 g accuracy. The biofilter system utilized plastic balls as filter media, and water was sterilized using a UV lamp installed on the inflow branch of the biofilter tower. A drum filter was used to remove waste and uneaten feed. The daily water exchange rate was 6–7%, while daily feed rations varied between 1% and 2.5% of the fish biomass weight, depending on the water temperature, fish growth rate, and feeding response. Throughout the experiment, the fish were fed with home-produced, coarse-grained feed (3–4.5 mm) containing 36% protein and 8% fat. With an average mortality rate of 20%, this phase resulted in the production of 1443 carp fingerlings with a total biomass of 173 kg per production cycle. No fish were sold at this stage; instead, all the biomass produced during phase 2 was transferred to phase 3.

After phase 2, the fish were transferred at 120 g to the final phase of production. The average survival rate during this phase was 97%, resulting in 1400 market-sized common carp at the end of production. This phase lasted 35 weeks, during which the fish reached a market weight of 3 kg per individual. The fish production system in this phase had a total usable water volume of 84 m^3^, consisting of seven polypropylene circular tanks, each with a capacity of 12 m^3^. Oxygen saturation was maintained between 70% and 80% through atmospheric circulation, while the water temperature was stabilized between 20 °C and 22 °C. The dissolved oxygen levels (mg/liter, %) in the water and the water temperature were monitored daily using a HACH HQ30d portable meter. Daily feed rations varied by around 2% of the fish biomass weight, depending on the water temperature, fish growth rate, and real-time feeding behavior. Feed was distributed twice daily using an automatic feeder (Linn Profi Automatic Feeder, 5 kg capacity). During phase 3, the population density in the tanks was 2.1 kg/m^3^, increasing to a final biomass density of 50 kg/m^3^ at the end of production. Feed consumption in this phase was approximately 71.4 kg/m^3^.

Financial data were collected for 2024, including input and output prices, as well as the average values of cost items for the year (Table 2). Data originally recorded in Hungarian forint (HUF) were converted into euros (EUR) based on the 2024 average exchange rate of HUF 395.2/EUR, as published by the Hungarian National Bank [36].

To supplement the primary analysis, the international and Hungarian literature related to the topic was reviewed using Scopus, Web of Science, and Google Scholar. The literature sources were analyzed using analytical methods. Additionally, data from relevant international (FAO—Food and Agricultural Organization of the United Nations, EUROSTAT—Statistical Office of the European Communities) and Hungarian statistical databases (HCSO—Hungarian Central Statistical Office, IAE—Institute of Agricultural Economics) were processed for this study.

### 2.2. Data Analysis

A deterministic economic model was developed following the validation of the collected data. Simulation modeling is the most appropriate method for determining the profitability of farm-level production [37]. The developed deterministic model follows a cost–benefit analysis framework that predicts outcomes based on fixed inputs, i.e., determining economic feasibility under different scenarios. Similar deterministic modeling has been used by several researchers [38,39,40,41] to investigate the economic value of key factors affecting the profitability of various production activities in livestock science. Szőllősi and Szűcs [42] applied deterministic models for the economic analysis of slaughter chicken production, while Feketéné Ferenczi et al. [43] used these models to examine the possibilities of economically sustainable honey production. Cehla et al. [44] used deterministic models to evaluate the economic feasibility of sheep production pathways. Additionally, Varga et al. [45] explored the impact of changing environmental conditions on ecosystem-based pond aquaculture using these models.

In line with the work of Engle [46], Engle et al. [47], and Dantas et al. [48], our model includes the budget of the entire activity, incorporating the principles outlined in Table 3. The key characteristic of the technology is that breeding can be reproduced at any time, allowing for the generalization of fundamental technological data in economic calculations. The model does not derive costs and revenues from analytical records and accounting data; instead, it assigns prices to the inputs based on technological and production data. The detailed costs of hatchery operations are not included in the economic model, but the purchase of larvae is accounted for at market prices.

Variable costs include the purchase of the breeding material (larvae), commercial feed, electricity, water, and other costs, such as machinery, veterinary and other external services, and consumables (e.g., rubber gloves, plankton nets). Fixed costs consist of labor, equipment, and facilities maintenance and depreciation. The value of fixed assets is accounted for annually through depreciation, which is calculated on a straight-line basis over the length of each production phase. In connection with the operation of fixed assets, in addition to depreciation, the costs of repairs and maintenance must be taken into account.

The economic model was complemented by sensitivity analyses (cross-table analysis) to assess how changes in key parameters affect profit [8,49]. The impacts of changes in the sales price, the feed price, specific gross yield, the electricity price, and electricity consumption on cost and profit were examined. In the cross table, feed and electricity prices were set at the extremes of price fluctuations over the past three years. For the sales price, the minimum extreme was based on the average price in 2022, while the maximum was estimated based on the rate of increase in previous years. The two extreme values for electricity consumption were determined by a +/−10% variation. For the yield, the extremes were defined by the specificity of the technology and capacity utilization.

It is important to point out that the economic model considers only costs and revenues related to the activity and does not include general expenses. This methodological approach can also be applied to other fish farming systems, particularly those aimed at enhancing economic sustainability. For example, Mihály-Karnai et al. [50] conducted a comprehensive analysis of production efficiency by combining a closed nursery system with a grow-out pond for common carp, enabling market-sized carp to be produced within two years [50]. Based on the results, decisions can be made that maximize production efficiency and reduce costs.

## 3. Results

Fish production in a RAS is a state-of-the-art industrial activity, in which all production phases are continuously monitored with minimal or no influence from natural processes. This enables a homogeneous commodity base and highly efficient production. In Hungary, the expansion of pond common carp production is constrained by the limited availability of pond space. Therefore, a potential breakthrough for increasing yields could be the adoption of closed, intensive fish production, particularly in fry production.

### 3.1. Inputs and Production Costs of Common Carp Production in RASs

When analyzing the different production phases separately, significant differences in production costs and the cost structure were observed (Table 4). In phase 1, the total production costs for intensive farming amount to EUR 651.75, which corresponds to 0.35 EUR/ha. The variable costs account for 11.87% of the total costs. The most important factor is the energy cost (40.51%) due to high electricity consumption (2.40 kWh/day). Other key variable costs include the cost of the purchased breeding material (30.42%) and feed costs (16.56%). The latter is attributed to the use of relatively expensive live and rearing feeds, despite lower overall feed consumption. The use of high-quality feed contributed to relatively rapid growth. Higher consumption was observed with the larger size of special starter feeds. Small quantities of *Artemia* spp. and BioMar Inicio Plus G were introduced to help the fish adapt to the artificial feed, while larger amounts of BioMar Inicio Plus 0.5 and 0.8 were used. Water and other costs represent a small proportion of the total costs, as larvae can be reared at sufficient population densities in a single 0.35 m^3^ tub. The majority of costs (88.13%) are fixed costs. The labor costs exceed the depreciation and maintenance costs, indicating that intensive systems require substantial labor input due to the constant care and professional handling needed for fish. Staff are responsible for maintaining fish health, ensuring strict water quality control, and observing the proper functioning of the technological system. Feeding the fish during different growth stages and monitoring their physiological condition are critical tasks that require specialized expertise. This high professional requirement contributes to increased labor costs in intensive aquaculture systems. In phase 1 (larval rearing and pre-rearing), the labor requirements are relatively high, accounting for 99.12% of the fixed costs. This is because managing early-stage fry is more challenging and requires increased inputs, particularly for live food feeding (*Artemia* spp.). Initially, feeding is conducted manually every three hours. However, as the fish grow, feeding can be automated using automated feeders, reducing the need for manual labor. This cost structure allows for the flexible scaling of production since the fixed costs remain constant.

Phase 2 involves rearing fish at a lower population density than in the previous phase, with a production cost (excluding the value of the fish produced in phase 1) of EUR 2869.66. The variable costs account for 76.63% of the total costs, with energy and feed costs being the most significant factors. Electricity consumption for the plant under study was 7992 kWh, making energy costs particularly high, representing 79.08% of the variable costs. In phase 2, feed consumption increases nearly tenfold (from 2.1 kg to 205 kg); however, the feed itself is significantly cheaper (1.1 EUR/kg) as the grain size increases. The fixed costs account for 23.37% of total production costs, with labor being the largest component, accounting for 84.91% of the fixed costs.

The market-sized fish production phase constitutes nearly 65% of the total production time, underscoring its crucial role in cost management. The high proportion of variable costs (91.41%) compared to previous phases indicates that a significant proportion of the production costs is directly linked to materials and energy usage. This is primarily due to the fact that daily electricity consumption more than doubles compared to the previous phase, as the fish are reared in larger tanks. Total electricity consumption in this phase is 41,472 kWh. Feed costs are also considerable, accounting for 29.90% of the variable costs, which corresponds to the feeding of nearly 6000 kg of feed. Notably, 96.1% of the total feed consumption for the entire production cycle occurs in this third phase. Water costs are also substantial, accounting for 19.04% of the variable costs due to significant water usage. The fixed costs represent 8.59% of the total costs, with labor being a dominant component, making up 90.91% of the fixed costs.

The following section presents an aggregated analysis of the production costs for the entire production cycle. Table 5 details the production costs by cost item at the plant level and per kilogram of market-sized common carp. The total production cost across all three phases amounts to EUR 22.964.71 per year, equating to EUR 5.47 per kilogram of market-sized common carp. Due to the specific nature of the technology, the most significant cost components are energy (47.01%) and feed (24.18%), which together account for 71.19% of the total production costs.

### 3.2. Profitability of Common Carp Production in RASs

Table 6 presents the sales revenue and profit generated by the activity. The sales revenue was derived from the sale of market-sized fish based on 2024 prices. A total fish yield of 4.2 tonnes can be achieved for the entire production cycle, amounting to EUR 21,255.06. While the sales revenue covers the variable costs, it does not fully offset the fixed costs. Consequently, under the economic conditions of 2024 and for the plant size considered, a loss of EUR 1709.65 was realizable. This corresponds to a loss of EUR 0.41 per kilogram of market-sized common carp.

The financial efficiency indicators (Table 7) are also unfavorable. The return on sales is −8.04%, meaning that the activity generates a loss of EUR 8.04 per EUR 100 of sales revenue. Regarding the profit-to-cost ratio, for every EUR 100 of production cost, a loss of EUR 7.44 is incurred under 2024 price conditions. Additionally, efficiency indicators per hour worked show that employees generate EUR 50.61 in sales revenue per hour.

The unit cost is high due to the substantial production costs. The cost of producing a market-sized common carp exceeds its sales price. This discrepancy is largely attributable to the employed technology: the fish are fed exclusively with compound feed (eliminating natural food sources), the mechanized system requires considerable electricity, and the rearing process demands a high level of professional expertise.

### 3.3. Sensitivity Analysis of the Financial Results

The effects of changes in the feed purchase price, the market-sized common carp sales price, specific gross yield, the electricity price, and electricity consumption on costs and profit were examined. These variables are the primary determinants of the evolution of cost and profit trends in common carp production within a RAS.

Table 8 clearly shows that the variation in the specific gross yield of common carp and the average feed price have a significant impact on unit cost. Based on the extreme values of the examined factors, the unit cost decreases by up to EUR 2. In contrast, for a given yield level, an increase of 0.01 EUR/kg in the feed price raises the cost price by 0.02 EUR/kg. At a given average feed price, the unit cost declines as yield increases, although at a diminishing rate.

Table 9 illustrates the impact of changes in the sales price of market-sized common carp and the average feed price on profit. At a feed price of EUR 0.85/kg, a profit improvement of EUR 2.00/kg is observed between the two extremes of the sales price. The difference in profit between the best (high sales price and low feed price) and worst (low sales price and high feed price) combinations of these two factors is 2.12 EUR/kg. A 0.01 EUR/kg increase in the feed price of a given sales price reduces the profit by 0.015 EUR/kg. At the 2024 market feed price (0.89 EUR/kg), a minimum sales price of EUR 5.47/kg is required to cover production costs. However, at a lower feed price (0.85 EUR/kg), the minimum sales price decreases to EUR 5.41/kg.

Due to the specific nature of the technology, the energy cost represents one of the most significant costs. Consequently, the impact of the change in both the sales price and the electricity price on profit was examined (Table 10). For a given sales price, an increase of 0.01 EUR/kWh in the electricity price reduces the profit by 0.12 EUR/kg. Between the two extremes—a high sales price and a low electricity price versus a low sales price and a high electricity price—the difference in profit is 3.06 EUR/kg. Under the electricity prices calculated for 2024, the production costs are covered if the sales price is 5.47 EUR/kg. Conversely, given the 2024 sales price, a maximum electricity price of 0.183 EUR/kWh is required to avoid losses.

It was also important to examine how changes in electricity use and electricity prices affect the unit cost (Table 11). Based on the electricity consumption calculated in the baseline model, an increase of 0.01 EUR/kg in the electricity price results in an increase of 0.12 EUR/kg in the unit cost. Under the electricity prices calculated for 2024, a 10% reduction in electricity consumption would lower the unit cost by 0.260 EUR/kg, while at higher electricity prices, the reduction could reach 0.283 EUR/kg. A minimum of 15.8% in electricity savings is required to cover the production costs calculated under 2024 economic conditions.

## 4. Discussion, Recommendations, and Limitations

Intensifying production in RASs increases the fish biomass per unit of breeding volume and enhances growth rates through the use of compounded feed and advanced technological equipment. This intensive farming approach not only demonstrates the technological efficiency of RASs but also significantly influences economic outcomes by reducing production times. Fish in RASs exhibit lower mortality rates and improved feed conversion ratios, both of which directly contribute to cost reductions.

Several studies [51,52,53,54,55,56] have confirmed that carp raised in RASs grow faster than those in traditional pond systems due to the continuous monitoring of environmental parameters such as water temperature and oxygen levels. Optimized growth conditions minimize stress and promote faster development. In our study, growth rates in a RAS were recorded at 0.02 g/day in phase 1, 1.32 g/day in phase 2, and 12.00 g/day in phase 3, with an overall average of 7.69 g/day across all phases.

The ability of RASs to produce market-sized fish (3 kg) within a year, coupled with the flexibility to stagger production cycles year round, enhances continuous fish production and optimizes resource utilization. However, the benefits of accelerated growth and year-round production in RASs come at the cost of higher feed and energy consumption compared to traditional three-year pond-based systems. While a RAS enables the rapid production of market-sized fish through high stocking densities and controlled environmental conditions, its increased resource demand—particularly for feed and energy—is considerably higher than that of extensive pond-based systems with longer and less controlled growth cycles [51,52]. This trade-off between increased efficiency and higher resource use is a critical factor in RAS operations, underscoring the need for careful management to balance productivity with long-term sustainability [21,24,57,58,59,60].

The cost structure of intensive carp farming is dominated by variable costs, which account for 87.31% of the total production costs, with electricity (53.84%) and feed (27.69%) being the most significant cost components. Labor costs represent nearly 12% of the total production costs, with the highest labor demand occurring in the early production phase due to the need for skilled management and monitoring. As production progresses, the integration of automated feeding systems gradually reduces manual labor requirements.

Despite their efficiency in controlled production, RASs remain financially challenging under current market conditions. In 2024, the unit production cost of market-sized common carp in a RAS was 5.47 EUR/kg, while the market price reached only 5.06 EUR/kg, resulting in a loss of 0.41 EUR/kg. This highlights the critical role of market pricing as a limiting factor in intensive common carp farming [61,62,63]. Nævdal [64] demonstrated that price fluctuations affect the volume harvested and the level of processing [64]. Common carp also struggle to compete with higher-value species in terms of price and consumer demand, further constraining the profitability of RASs. Although the sales price of market-sized common carp from pond production has steadily increased in recent years [65], reaching 5.06 EUR/kg, the highest price ever recorded in Hungary, these prices remain insufficient to cover the higher investment and maintenance costs associated with more intensive systems [60,66].

RASs provide numerous sustainability benefits, including controlled environmental conditions, year-round production cycles, the consistent rearing of both market-sized fish and larvae, and improved biosecurity [21]. Nonetheless, these systems can only be economically viable if high population densities are maintained and if sales prices exceed current market prices [64,67,68]. If the upward trend in sales prices continues, coupled with only modest increases in operating costs, intensive carp production could become profitable.

It is important to note that the calculated costs were assessed on an activity basis rather than at the enterprise level, including those incurred by small-scale operations. General expenses may increase the total production costs of the business. Nevertheless, increasing farm size can yield economies of scale that reduce unit cost and improve profitability [8] and enhance the sustainability of the sector by reducing environmental impacts while boosting economic viability [60,66]. In the future, factors influencing profitability must include the additional expenditure associated with achieving sustainability, which is required to comply with ESG-mandated (environmental, social, and governance) standards. A significant challenge is that these costs are expected to primarily impact producers in the agricultural sector. The integration of ESG considerations will impose significant additional costs on primary producers, as they will need to adapt their operations, supply chains, and reporting processes to meet increasingly stringent regulatory and market expectations [69].

To lower high energy costs, it is important to integrate energy-use measures into operations, as Presa et al. [8] draw attention to. Implementing these solutions can help reduce energy consumption and operating costs in the long term, thereby enhancing production efficiency and sustainability. This includes modernizing water circulation and oxygen supply systems, which consume a significant amount of electricity. It is advisable to use motor pumps and motors that meet current frequency requirements for optimal performance. Automated and intelligent energy management systems designed to control temperature and oxygen levels and to incorporate geothermal energy can lead to substantial savings. To lessen reliance on traditional energy sources, renewable options such as solar panels and heat pumps should be considered, as these can power the facility itself. By installing solar panels, aquaculture facilities can meet a significant portion of their electricity demands, thereby reducing reliance on grid electricity and lowering greenhouse gas emissions, similar to trends observed in other sectors [70,71,72,73]. Sustainable solutions are not only environmentally friendly but also financially advantageous. Innovative technologies can minimize or reduce electricity consumption, leading to lower production costs [74,75]. This is further supported by the observation that a minimum of 15.8% in electricity savings is required to cover the production costs calculated under 2024 economic conditions. The implementation of sustainability-driven initiatives will likely require additional investments in green energy solutions, waste management technologies, and supply chain transparency, further impacting financial feasibility.

It is important to note that, for the improvement of profitability, the choice of energy source has a significant impact on both economic and ecological indices in RASs. Fossil fuel-based energy increases operational costs and volatility, thereby reducing profitability, while alternative energy sources, particularly solar energy, provide a more sustainable solution by enhancing energy efficiency and reducing dependence on fossil fuels [72,73]. Solar energy is particularly promising, as installation costs continue to decrease while technological efficiency improves. Although the initial investment in photovoltaic (PV) systems may be high, the long-term benefits—such as lower electricity costs, reduced carbon emissions, and greater resilience against market price fluctuations—make it an attractive alternative [70]. Furthermore, integrating solar energy with energy storage solutions, such as battery systems, can ensure a continuous power supply, mitigate potential operational disruptions, and contribute to the long-term profitability and sustainability of RAS operations [71].

Feed costs are important, but effective feed utilization deserves special attention in the production process [76]. Maintaining feed costs at an appropriate level is crucial for economical and sustainable common carp farming, and management is another critical aspect of financial sustainability. Given current market conditions and the growing variety of available feeds, it would be beneficial to develop carp-specific feeds that offer the optimal nutritional composition for the fish in a more cost-effective and environmentally friendly manner [76,77,78]. The use of alternative protein sources, such as plant-based and insect-derived feeds, can reduce dependency on traditional fishmeal, supporting both economic and environmental sustainability. Maintaining efficient feeding strategies and adjusting feeding schedules based on growth phases can further optimize feed utilization and minimize waste [16,17]. Under 2024 feed prices, a sales price of at least 5.47 EUR/kg is necessary for producers to achieve a profit. However, at the lowest feed price, this requirement drops slightly to 5.41 EUR/kg. Therefore, promoting scientific research and fostering industrial cooperation to ensure that producers have access to high-quality, competitive feed is advantageous.

Reducing labor and maintenance costs is essential for enhancing profitability and maintaining operational efficiency. Continuous production cycles, where different age groups of fish (juvenile, pre-farmed, and market-sized) are cultivated simultaneously, help balance the labor demand throughout the year and maximize infrastructure utilization [14]. This approach, combined with optimized maintenance strategies, reduces costs and enhances production stability, efficiency, and long-term sustainability [10]. Continuous production facilitates a more consistent utilization of labor and infrastructure, which helps to avoid peak loads and prolonged downtimes. This approach enables continuous market production, better aligning supply with consumer demand [9].

Our research has shown that the shortening of the growing season combined with increased production intensity leads to higher unit costs for market-sized common carp. This is primarily due to the higher feed costs and greater electricity requirements associated with intensified farming technologies. Common carp production in a one-year RAS was unprofitable under the economic conditions of 2024. It also has a comparative disadvantage compared to traditional three-year pond fish production and a combined two-year strategy [50,79]. However, this approach could represent a breakthrough opportunity and could be economically beneficial for carp farmers in the future. Using a RAS can conserve the resources needed for pond maintenance, fuel, and fish stocking while also addressing issues with piscivorous predators and water shortages stemming from climate change [33].

The primary limitation of this study is the experiment’s time frame and lack of repetitions, as a one-year period may not fully capture long-term economic and environmental fluctuations. Key factors such as feed prices, electricity costs, and market demand vary over time, potentially impacting the profitability and sustainability of RASs. This study presents a cross-sectional economic analysis with findings specific to the 2024 economic year. However, the sensitivity (cross-table) analysis clearly illustrates the economic effects of price changes. Additionally, seasonal variations, such as water temperature fluctuations and disease outbreaks, may be underrepresented in a short-term study, despite their potential impact on production efficiency. Climate change further complicates these challenges by altering the water temperature and quality, which can increase energy demands for cooling, affect fish growth rates, and raise operational costs. These combined factors may disrupt production processes, ultimately influencing the economic feasibility and environmental sustainability of RAS operations.

Future research should focus on market trends, efficiency improvements, and policy frameworks that facilitate the transition to sustainable and economically viable RAS production. Moreover, a life cycle assessment (LCA) comparing solar energy with petroleum-based energy would offer valuable insights into sustainability. By addressing these challenges, the economic sustainability of intensive common carp farming can be significantly improved.

## 5. Conclusions

This study assessed the economic feasibility of experimentally reared common carp in a small-scale RAS farm under the economic conditions of 2024 in Hungary. The findings indicate that while RASs offer advantages in controlled production, biosecurity, and environmental stability, their financial viability is constrained by high feed and energy costs. The results show that market-sized common carp (3 kg each) can be produced within a year under intensive farming conditions. High stocking densities and precise environmental control accelerate growth and reduce mortality.

However, these benefits come at the expense of significantly higher feed and energy consumption than traditional three-year pond-based production. The economic analysis revealed that the unit cost of market-sized common carp was 5.47 EUR/kg in 2024, while the average sales price was 5.06 EUR/kg, resulting in a loss of EUR 0.41 per kilogram. This highlights the substantial gap between production costs and market returns, underscoring the financial challenges associated with RAS-based carp farming. The cost structure reveals that variable costs accounted for 87.31% of the total production costs, with electricity (53.84%) and feed (27.69%) being the two most significant cost components. This highlights the critical areas where efficiency improvements could lead to cost reductions and, potentially, improved profitability. Labor costs represented nearly 12% of the total production costs, primarily due to a higher labor demand in phase 1, which gradually declined as automated feeding systems reduced the manual input in later production phases.

Sensitivity analyses confirmed that the economic performance of RAS-based carp farming is highly sensitive to key cost drivers. A 0.01 EUR/kg increase in feed prices raised unit costs by 0.015 EUR/kg, while a 0.01 EUR/kWh rise in electricity prices led to a 0.12 EUR/kg increase in production costs, emphasizing the substantial impact of energy consumption. Market price fluctuations also significantly influenced profitability, highlighting the importance of achieving higher sales prices or reducing operational costs. Additionally, improvements in specific gross yield were identified as a critical factor in reducing unit costs, as higher yields contribute directly to enhanced cost efficiency. Under the economic conditions of 2024, RAS-based common carp farming was not financially viable. However, the findings suggest that improving cost efficiency, optimizing production processes, and achieving higher market prices could enhance economic feasibility in the future.

## Figures and Tables

**Figure 1 animals-15-01055-f001:**
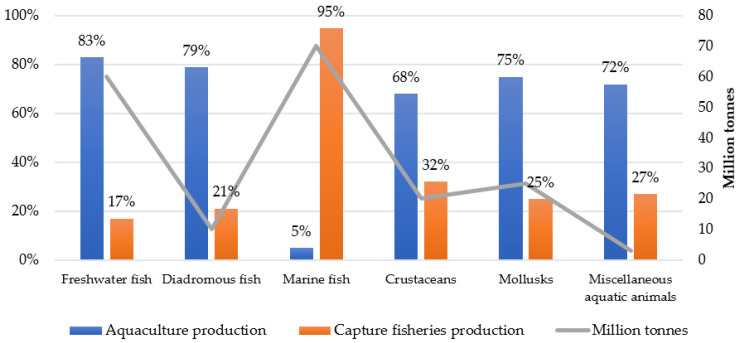
Production of aquatic animals by ISSCAAP division. Source: Own editing (2025) based on [6]. The International Standard Statistical Classification of Aquatic Animals and Plants, abbreviated as ISSCAAP, is a nomenclature developed by the United Nations (UN) Food and Agriculture Organization (FAO).

**Figure 2 animals-15-01055-f002:**
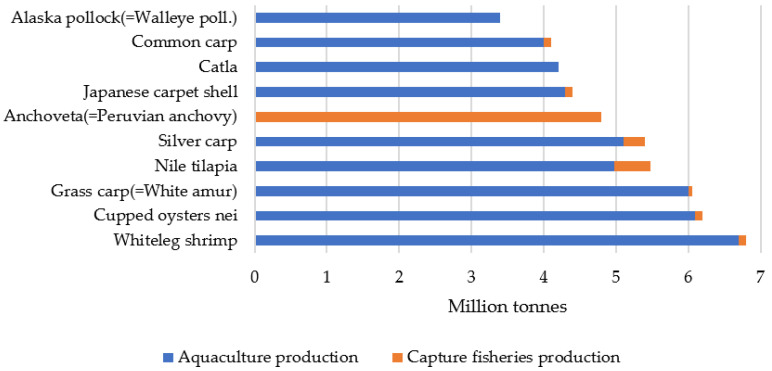
The top ten species in the world’s fisheries and aquaculture, 2024. Source: Own editing (2025) based on [6].

**Figure 3 animals-15-01055-f003:**
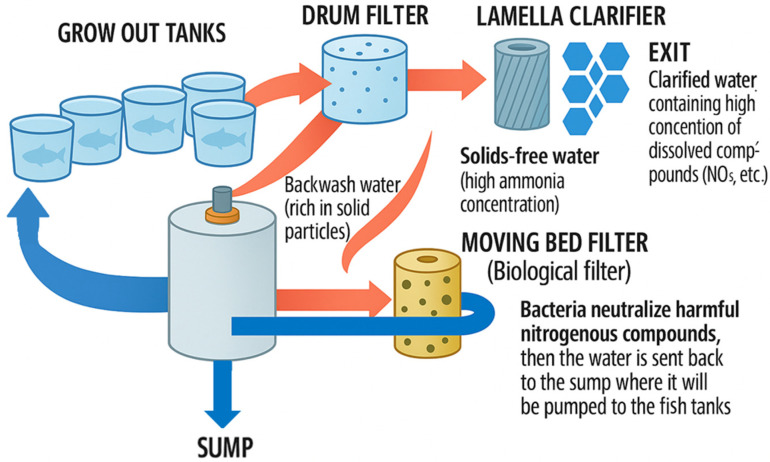
Schematic design of a recirculated aquaculture system (RAS) farm. Own editing based on [34].

**Table 1 animals-15-01055-t001:** Production data of common carp production under the conditions of a RAS production strategy. Source: Own editing and farm-level data (2021).

Description	Phase 1	Phase 2	Phase 3
	Pre-Rearing	Post-Rearing	Market-Sized Fish Production
Phase length (days/phase)	60	90	215
Number of units (pool)	1	12	7
Production capacity (m^3^)	0.35	12.00	84.00
Fish loss (%)	40	20	3
Time of stocking (month)	February	April	July
Average body weight (g) of carp at the time of establishing the population	0.01	1.05	120
Stocking density (fish/m^3^)	8 857	150	17
Time of harvest (month)	April	July	February
Average body weight of carp at harvest (g)	1.05	120	3 000
Average carp growth rate (g/day)	0.02	1.32	12.00
Specific gross yield of carp (kg/m^3^)	5.58	14.43	50.00
Feed conversion ratio (FCR) (kg/kg)	1.05	1.19	1.43
Total feed conversion ratio (kg/kg)	1.48		
Electricity consumption (kW/m^3^)	0.286	0.308	0.086

**Table 2 animals-15-01055-t002:** Basic financial data: input and output prices (EUR 1 = HUF 395.2). Source: Own editing and farm-level data (2024).

Name of Phases	Description	Unit of Measure	Value
Phase 1	Cost of breeding material (larval)	EUR/pc	0.008
	Feed prices		
	*Artemia* spp./Brine Shrimp	EUR/kg	253.036
	Inicio Plus G (63/11)	EUR/kg	7.100
	Inicio Plus 0.5 (58/15)	EUR/kg	3.720
	Inicio Plus 0.8 (56/18)	EUR/kg	3.600
	Electricity price	EUR/kWh	0.218
	Water cost	EUR/m^3^	4.127
	Other costs *	EUR/m^3^	14.459
	Wage (worker)	EUR/h	6.326
	Depreciation and maintenance	EUR/m^3^	14.459
Phase 2	Feed prices (Aqua Start 1.2–2 mm)	EUR/kg	1.100
	Electricity price	EUR/kWh	0.218
	Other costs *	EUR/m^3^	4.217
	Wage (worker)	EUR/h	6.326
	Depreciation and maintenance	EUR/m^3^	8.435
Phase 3	Feed prices (Aqua Garant Classic mix 2–6 mm)	EUR/kg	0.886
	Electricity price	EUR/kWh	0.218
	Other costs *	EUR/m^3^	0.602
	Wage (worker)	EUR/h	6.326
	Depreciation and maintenance	EUR/m^3^	1.807
	Sales price of market-sized common carp **	EUR/kg	5.061

* Other costs include fuel consumption, the servicing of the machines used in the activity, the use of animal health and other external services, and consumables (rubber gloves, fishing nets, plankton nets etc.). ** The ex-farm price is the net price per kilogram of the first sale of the marketed fish produced net of transport costs.

**Table 3 animals-15-01055-t003:** The mathematical correlations of the deterministic economic model. Source: Own editing (2025).

Description	Correlation
Sales revenue	market-sized fish (kg) × sales price of market-sized fish (EUR/kg)
Total variable costs	$\Sigma_{i=1}^{6}\;k_{i}$
Cost of breeding material (k_1_)	average price of breeding material (EUR/fish) × stocking density (number of fish/m^3^) × production capacity (m^3^)
Energy cost (k_2_)	electricity used (kW/m^3^) × production capacity (m^3^) × 24 (h) × production lengh (day) × electricity price (EUR/kWh)
Water cost (k_3_)	cost of water used (EUR/m^3^) × production capacity (m^3^)
Feed cost (k_4_)	amount of feed applied to the area (kg/m^3^) × production capacity (m^3^) × feed price (EUR/kg)
Other cost (k_6_)	value of other material costs (EUR/m^3^) × production capacity (m^3^)
Total fixed costs	k_7_ + k_8_
Labor costs (k_7_)	labor input (number of hours worked) × hourly rate (EUR/working hour)
Depreciation and maintenance (k_8_)	depreciation and cost of maintenance (EUR/m^3^) × production capacity (m^3^)

**Table 4 animals-15-01055-t004:** Costs of the three different phases of the intensive common carp production strategy (2024) (Unit: EUR). Source: Own data collection and calculation (2025).

Description	Phase 1	Phase 2	Phase 3
Breeding material *	23.53	-	-
Energy	31.34	1739.15	9024.78
Water	4.62	183.24	3384.16
Feed	12.81	226.12	5313.71
Other	5.06	50.61	50.61
Total variable costs	77.36	2199.12	17,773.26
Labor	569.33	569.33	1518.22
Depreciation and maintenance	5.06	101.21	151.82
Total fixed costs	574.39	670.54	1670.04
Total production costs	651.75	2869.66	19,443.30

* Only the first phase includes the cost of purchasing breeding material; the other phases include only the costs without breeding fish, i.e., the costs of further rearing.

**Table 5 animals-15-01055-t005:** Production costs for the entire production process in the intensive common carp production strategy (2024). Source: Own data collection and calculation (2025).

Description	Value (EUR)	Value per 1 kg of Market-Sized Common Carp (EUR/kg)	Distribution (%)
Breeding material	23.53	0.01	0.10
Energy	10,795.27	2.57	47.01
Water	3572.02	0.85	15.56
Feed	5552.64	1.32	24.18
Other	106.28	0.03	0.46
Total variable costs	20,049.74	4.78	87.31
Labor	2656.88	0.63	11.57
Depreciation and maintenance	258.09	0.06	1.12
Total fixed costs	2914.98	0.69	12.69
Total production costs	22,964.71	5.47	100.00

**Table 6 animals-15-01055-t006:** Costs and profit of the intensive common carp production strategy (2024). Source: Own data collection and calculation (2025).

Description	Value (EUR)	Value per 1 kg of Market-Sized Common Carp (EUR/kg)	Value per 1 m^3^ * (EUR/m^3^)
Sales revenue	21,255.06	5.06	220.60
Total production costs	22,964.71	5.47	238.35
Profit	−1709.65	−0.41	−17.75

* These values are compared to the total capacity of the entire production (96.35 m^3^).

**Table 7 animals-15-01055-t007:** Financial efficiency in the intensive common carp production strategy (2024). Source: Own data collection and calculation (2025).

Description	Value	Unit
Return on sales (profit/sales revenue × 100)	−8.04	%
Profit-to-cost ratio (profit/total production cost × 100)	−7.44	%
Sales revenue per one hour worked	50.61	EUR/h
Profit per one hour worked	−4.07	EUR/h
Unit cost of 1.2 g of offspring (after phase 1)	333.72	EUR/kg
Unit cost of 120 g of carp (after phase 2)	20.22	EUR/kg
Unit cost of market-sized common carp (after phase 3)	5.47	EUR/kg

**Table 8 animals-15-01055-t008:** Estimated unit cost (EUR/kg) in the intensive common carp production strategy as a function of the specific gross yield of common carp and the average feed price (2024). Source: Own data collection and calculation (2025).

		Specific Gross Yield of Common Carp (kg/m^3^)									
		42	44	46	48	50	52	54	56	58	60
**Average feed price** (EUR/kg) *	**0.85**	6.431	6.139	5.872	5.627	5.402	5.194	5.002	4.823	4.657	4.502
	**0.86**	6.449	6.155	5.888	5.643	5.417	5.208	5.016	4.836	4.670	4.514
	**0.87**	6.466	6.172	5.904	5.658	5.432	5.223	5.029	4.850	4.682	4.526
	**0.88**	6.484	6.189	5.920	5.673	5.446	5.237	5.043	4.863	4.695	4.539
	**0.89**	6.501	6.206	5.936	5.689	5.461	5.251	5.057	4.876	4.708	4.551
	**0.90**	6.519	6.223	5.952	5.704	5.476	5.265	5.070	4.889	4.721	4.563
	**0.91**	6.537	6.239	5.968	5.719	5.491	5.280	5.084	4.902	4.733	4.576
	**0.92**	6.554	6.256	5.984	5.735	5.505	5.294	5.098	4.916	4.746	4.588
	**0.93**	6.572	6.273	6.000	5.750	5.520	5.308	5.111	4.929	4.759	4.600
	**0.94**	6.589	6.290	6.016	5.766	5.535	5.322	5.125	4.942	4.772	4.613

* The average feed price is EUR 0.894 in the basic model, which is the weighted mean of the price and quantity of feed used in each phase of production.

**Table 9 animals-15-01055-t009:** Estimated profit (EUR/kg) for the intensive common carp production strategy as a function of the sales price and the average feed price (2024). Source: Own data collection and calculation (2025).

		Sales Price of Market-Sized Common Carp (EUR/kg)								
		3.50	3.75	4.00	4.25	4.50	4.75	5.00	5.25	5.50
**Average feed price** (EUR/kg) *	**0.85**	−1.902	−1.652	−1.402	−1.152	−0.902	−0.652	−0.402	−0.152	0.098
	**0.86**	−1.917	−1.667	−1.417	−1.167	−0.917	−0.667	−0.417	−0.167	0.083
	**0.87**	−1.932	−1.682	−1.432	−1.182	−0.932	−0.682	−0.432	−0.182	0.068
	**0.88**	−1.946	−1.696	−1.446	−1.196	−0.946	−0.696	−0.446	−0.196	0.054
	**0.89**	−1.961	−1.711	−1.461	−1.211	−0.961	−0.711	−0.461	−0.211	0.039
	**0.90**	−1.976	−1.726	−1.476	−1.226	−0.976	−0.726	−0.476	−0.226	0.024
	**0.91**	−1.991	−1.741	−1.491	−1.241	−0.991	−0.741	−0.491	−0.241	0.009
	**0.92**	−2.005	−1.755	−1.505	−1.255	−1.005	−0.755	−0.505	−0.255	−0.005
	**0.93**	−2.020	−1.770	−1.520	−1.270	−1.020	−0.770	−0.520	−0.270	−0.020
	**0.94**	−2.035	−1.785	−1.535	−1.285	−1.035	−0.785	−0.535	−0.285	−0.035

* The average feed price is EUR 0.894, which is the weighted mean of the price and the quantity of feed used in each phase of production.

**Table 10 animals-15-01055-t010:** Estimated profit (EUR/kg) in the intensive common carp production strategy as a function of sales price and electricity price. Source: Own data collection and calculation (2025).

		Sales Price of Market-Sized Common Carp (EUR/kg)								
		3.50	3.75	4.00	4.25	4.50	4.75	5.00	5.25	5.50
**Electricity price** (EUR/kWh)	**0.15**	−1.169	−0.919	−0.669	−0.419	−0.169	0.081	0.331	0.581	0.831
	**0.16**	−1.287	−1.037	−0.787	−0.537	−0.287	−0.037	0.213	0.463	0.713
	**0.17**	−1.405	−1.155	−0.905	−0.655	−0.405	−0.155	0.095	0.345	0.595
	**0.18**	−1.524	−1.274	−1.024	−0.774	−0.524	−0.274	−0.024	0.226	0.476
	**0.19**	−1.642	−1.392	−1.142	−0.892	−0.642	−0.392	−0.142	0.108	0.358
	**0.20**	−1.760	−1.510	−1.260	−1.010	−0.760	−0.510	−0.260	−0.010	0.240
	**0.21**	−1.878	−1.628	−1.378	−1.128	−0.878	−0.628	−0.378	−0.128	0.122
	**0.22**	−1.996	−1.746	−1.496	−1.246	−0.996	−0.746	−0.496	−0.246	0.004
	**0.23**	−2.114	−1.864	−1.614	−1.364	−1.114	−0.864	−0.614	−0.364	−0.114
	**0.24**	−2.232	−1.982	−1.732	−1.482	−1.232	−0.982	−0.732	−0.482	−0.232

**Table 11 animals-15-01055-t011:** Estimated unit cost (EUR/kg) in the intensive common carp production strategy as a function of electricity consumption and electricity price.

		Electricity Consumption								
		−10.0%	−7.5%	−5.0%	−2.5%	0%	+2.5%	+5.0%	+7.5%	+10.0%
**Electricity price** (EUR/kWh)	**0.15**	4.492	4.536	4.581	4.625	4.669	4.713	4.758	4.802	4.846
	**0.16**	4.598	4.646	4.693	4.740	4.787	4.835	4.882	4.929	4.976
	**0.17**	4.705	4.755	4.805	4.855	4.905	4.956	5.006	5.056	5.106
	**0.18**	4.811	4.864	4.917	4.970	5.024	5.077	5.130	5.183	5.236
	**0.19**	4.917	4.973	5.029	5.086	5.142	5.198	5.254	5.310	5.366
	**0.20**	5.024	5.083	5.142	5.201	5.260	5.319	5.378	5.437	5.496
	**0.21**	5.130	5.192	5.254	5.316	5.378	5.440	5.502	5.564	5.626
	**0.22**	5.236	5.301	5.366	5.431	5.496	5.561	5.626	5.691	5.756
	**0.23**	5.342	5.410	5.478	5.546	5.614	5.682	5.750	5.818	5.886
	**0.24**	5.449	5.520	5.590	5.661	5.732	5.803	5.874	5.945	6.016

## Data Availability

The original contributions presented in the study are included in the article/Appendix A, further inquiries can be directed to the corresponding author.

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
