# Peer review of "Sustainability in Intensive Aquaculture—Profitability of Common Carp (Cyprinus carpio) Production in Recirculating Aquaculture Systems Based on a Hungarian Case Study"

_animals, 2025, doi:10.3390/ani15071055_

Round 1

Reviewer 1 Report

Comments and Suggestions for Authors

The authors conducted very important research to help improve the production of the very promising common carp in RAS system with the data and conclusions obtained during the research.

Author Response

Dear Reviewer,

We very much appreciate the Reviewer’s valuable and constructive suggestions and comments for improving the manuscript. Below, we respond to each point and describe the corresponding modifications made to our study. Please find the revised manuscript with Tracked Changes showing the changes made.

Best regards,

Dániel Fróna 

Reviewer 2 Report

Comments and Suggestions for Authors

This study examines the economic feasibility and sustainability of intensive common carp (Cyprinus carpio) production in Recirculating Aquaculture Systems (RAS) in Hungary. It focuses on understanding the costs and profitability of using RAS for fish farming, specifically looking at factors like feed costs, electricity consumption, and market sale prices. The authors emphasize integrating economic, environmental, and technological considerations to enhance sustainable aquaculture practices, they note that RAS production may be unprofitable under current economic conditions, highlighting the need for continued price adjustments and operational improvements.

While the study offers important insights into the economic dynamics of RAS for carp production, weaknesses in sample size, time frame, and a lack of broader context or environmental considerations might limit the generalizability and long-term applicability of its findings.  These are some areas which the authors could consider to improve:

  1. The terminology used is academically sound, but adding more precision in certain areas would further enhance clarity and depth.  "Deterministic model": This is a well-accepted term in research when discussing models that predict outcomes based on fixed inputs. However, providing a brief explanation of the model type or referring to a specific model (e.g., linear programming or cost-benefit analysis) might enhance clarity, especially for readers unfamiliar with the term.

  1. Opportunities for optimizing feed management and energy use: This is a good phrase, but it might be strengthened by specifying "energy efficiency" instead of just "energy use," which would be more specific and aligned with sustainability discussions.
  1. The study relies on data from a semi-industrial scale experiment conducted in Hungary. However, it does not mention the number of replicates used or the variability in the data across different trials. The absence of sufficient replicates could limit the robustness of the conclusions, especially when dealing with economic models and sensitivity analyses. Without multiple replicates or trials, it becomes difficult to account for potential outliers or random variation, which could influence the results.
  2. The data were collected over a period of one year (2020-2021), which, although reasonable for understanding certain aspects of RAS production, may not be sufficient to capture long-term trends or variations. For example, changes in feed prices, electricity costs, or market demand might fluctuate over a more extended period, and a year-long study may not adequately account for these long-term variables. Additionally, the impact of seasonal variations (e.g., water temperature fluctuations, disease outbreaks) could be underrepresented in a short time frame.
  3. Climate Change Considerations: Climate change can affect RAS by altering water temperatures and quality, which may increase energy demands for cooling or affect fish growth rates. These shifts could raise operational costs and disrupt production, impacting both the economic viability and environmental sustainability of RAS.
  4. Energy Sources and Life Cycle Assessment (LCA): The study focuses on energy consumption but doesn’t fully explore the implications of different energy sources. Currently, RAS systems rely on petroleum-based electricity, which has a high carbon footprint and fluctuating costs. A Life Cycle Assessment (LCA) comparing solar energy with petroleum-based energy would offer valuable insights into sustainability. Solar energy, despite its high initial investment, can reduce long-term operational costs and greenhouse gas emissions, making it a more ecologically friendly and economically stable option.
  5. Impact on Economic and Ecological Indices:The choice of energy source affects both economic and ecological indices. Petroleum-based energy increases operational costs and volatility, reducing profitability, while solar energy reduces emissions and operational costs in the long term, improving profitability and sustainability.

Author Response

(The authors gave the same response as above.)

Reviewer 3 Report

Comments and Suggestions for Authors

PLS See attachment

Author Response

(The authors gave the same response as above.)

Round 2

Reviewer 3 Report

Comments and Suggestions for Authors

The author has made point-to-point revisions according to the reviewer's suggestions, and I agree to publish it after suitable text editing.

Author Response

Dear Reviewer,
We greatly appreciate your suggestions and comments. We have revised the introduction, discussion, and conclusion to make them more objective, and we hope these modifications meet your expectations. Please find attached the revised manuscript with tracked changes highlighting the alterations. Additionally, we have included the edited version without tracked changes for your convenience.
Best regards,
Dániel Fróna
